# The Mechanism of bnAb Production and Its Application in Mutable Virus Broad-Spectrum Vaccines: Inspiration from HIV-1 Broad Neutralization Research

**DOI:** 10.3390/vaccines11071143

**Published:** 2023-06-25

**Authors:** Xinyu Zhang, Zehua Zhou

**Affiliations:** 1Research Center for Infectious Diseases, Tianjin University of Traditional Chinese Medicine, Tianjin 301617, China; zxy9992020@163.com; 2Institute for Biological Product Control, National Institutes for Food and Drug Control (NIFDC) and WHO Collaborating Center for Standardization and Evaluation of Biologicals, No. 31 Huatuo Street, Daxing District, Beijing 102629, China; 3College of Life Science, Jilin University, Changchun 130012, China

**Keywords:** broadly neutralizing antibodies, various viruses, vaccines

## Abstract

Elite controllers among HIV-1-infected individuals have demonstrated a stronger ability to control the viral load in their bodies. Scientists have isolated antibodies with strong neutralizing ability from these individuals, which can neutralize HIV-1 variations; these are known as broadly neutralizing antibodies. The nucleic acid of some viruses will constantly mutate during replication (such as SARS-CoV-2), which will reduce the protective ability of the corresponding vaccines. The immune escape caused by this mutation is the most severe challenge faced by humans in the battle against the virus. Therefore, developing broad-spectrum vaccines that can induce broadly neutralizing antibodies against various viruses and their mutated strains is the best way to combat virus mutations. Exploring the mechanism by which the human immune system produces broadly neutralizing antibodies and its induction strategies is crucial in the design process of broad-spectrum vaccines.

## 1. Introduction

Coronavirus disease 2019 (COVID-19) is prevalent worldwide. By May 2023, it had infected 760 million people and killed more than 6.9 million people [1]. Due to the lack of specific drugs, the development and use of SARS-CoV-2 vaccines has become the main means to control the epidemic. SARS-CoV-2 is a single-stranded positive-sense RNA virus, and its exposed spike protein (S protein) and receptor-binding domain (RBD) on the S protein are the main targets for vaccine design [2]. Although many countries are promoting the research and development process of COVID-19 vaccines, SARS-CoV-2 verification is emerging, one after another. The immune escapes caused by mutations of the virus have reduced the effectiveness of developed vaccines, which also makes the continued spread of COVID-19 a major public health problem worldwide.

There have been many successful vaccines in human history, such as for smallpox, which have been optimized and developed based on previous experience. People who have been infected with smallpox will not be infected again, which provides inspiration for people. Injecting the pus from people with mild smallpox into the nose or skin of others can protect them from future smallpox infections [3], but this method has certain side effects [4,5]. In 1976, Edward Jenner discovered that abscesses on cows can also provide protection and reduce side effects [6], marking the official birth of vaccination. However, traditional vaccines have encountered issues such as insufficient immune response duration and decreased neutralizing ability against verifications in the prevention of SARS-CoV-2, resulting in unsatisfactory preventive effects [7,8]. The SARS-CoV-2 pandemic has shown the public the limitations of traditional vaccines being unable to effectively respond to viral mutation escape, which is also the reason why preventive broad-spectrum vaccines have not been introduced for viruses such as HIV-1 and influenza. Influenza can still be maintained by vaccination once a year, due to the predictability of antigenicity drift of its epidemic strain [9]. The special and increasing genetic diversity of HIV-1 virus makes it particularly challenging to develop a usable HIV-1 vaccine [10]. The first vaccine tested in clinical trials generally used env as an antigen to trigger neutralizing antibody, but in the seven efficacy tests of the HIV-1 vaccine completed so far, except for one, the tests have not been successful (Table 1). The only trial that showed a reduction in HIV-1 transmission efficacy (31.2%) was the CRFAE trial of RV144 (NCT00223080) in the Thailand_ 01 Canary pox/GP120 vaccine [11,12]. This experiment indicates that high levels of antibodies and low levels of envelope (Env) protein-specific IgA binding to the HIV-1 variable ring 2 (V2) are associated with reduced transmission, and this guides the design of subsequent clinical trials [12]. However, in two Phase IIb/III clinical trials aimed at improving the RV144 trial, (HVTN) 702 (NCT02968849) [13] and HVTN 705 (NCT03060629) [14,15] did not show significant efficacy [15], indicating that the RV144 trial may not be a precursor to vaccine success.

In addition to drawing on previous experience, the design of broad-spectrum vaccines requires a profound understanding of the immune system’s working methods. However, there are still many blind spots in our current understanding of how the immune system responds to different types of infections and the immune mechanisms of successful vaccines, which hinders the development of broad-spectrum vaccines. In this review, we list the limiting factors for inducing bnAb in healthy individuals and review the attempts to induce bnAb strategies and their applications in broad-spectrum vaccine design, including various methods used by different researchers. We hope this review will make a small contribution to the design of broad-spectrum vaccines against mutable viruses.

## 2. Broadly Neutralizing Antibody

The concept of a broadly neutralizing antibody (bnAb), which refers to antibodies with a wide neutralization against variations, first came from the field of HIV-1; it refers to antibodies with a wide neutralization against different HIV-1 strains [28]. This concept is now also applied to other mutable viruses for antibodies that can still neutralize after virus mutations. BnAb typically recognizes conserved epitopes, or can resist amino acid mutations in some variable epitopes [29]. Inducing bnAbs has become the main target of preventive viral vaccines.

### 2.1. Introduction to bnAbs

Among HIV-1-infected individuals, there are some who do not progress for a long time. They can effectively control the virus in their bodies, even without taking treatment drugs. These individuals are referred to as elite controllers. BnAb is selected from elite control individuals, and is the main antibody that exerts antiviral effects in the body [30]. BnAbs can not only bind to antigens and exert antiviral effects by binding to antibodies, but can also participate in antiviral immune responses by exerting ADCC effects through IgG Fc fragments [31].

Research has found that bnAbs mainly targets conserved epitopes such as CD4 binding sites on the HIV-1 envelope (Env) [32], but it is difficult to induce corresponding broad-spectrum neutralizing antibodies using these epitopes as immunogens [33]. Therefore, some researchers have conducted research on the mechanism of bnAb production [34], attempting to understand the characteristics of bnAbs through immune reactions. Shortly after the isolation of the first batch of bnAbs, researchers noticed that these antibodies exhibited one or more unusual features, including high-frequency V (D) J mutations, the third complementary determinant region in the significantly extended heavy chain variable region (HCDR3), and self-reactions with human lipids and proteins [35]. Based on these findings, researchers hypothesize that the immune tolerance mechanism is not conducive to the production of bnAbs [36,37]. The initial support for this hypothesis comes from the strong binding of two types of bnAbs (including 2F5 and 4E10) to human autoantigens [36]. The mutation frequency of V (D) J in many bnAbs is as high as 30%. When the V region gene of the heavy chain and the light chain of the antibody undergoes intense somatic cell hypermutation (SHM) (Figure 1), it is also necessary to avoid tolerance and maintain the ability to neutralize HIV-1 [38].

### 2.2. Limitations on the Production of bnAbs

Researchers believe that bnAbs are produced by mature memory B cells of HIV-1-infected people who have not progressed for a long time [39,40,41]. Due to the particularity and complexity of bnAbs, understanding the selective promotion of bnAbs development by CD4+T and B cells can provide insights for the development of mature broad-spectrum HIV-1 vaccines. BnAb is one of the antibodies that has experienced the most SHM. Florian et al. found a significant decrease or complete loss of antibody affinity after reverse mutation to approach the inferred lineage sequence [42,43], indicating that SHM is necessary for the formation of bnAbs [44], and that SHM relies on the help of Tfh cells in the germinal center (GC). Therefore, scientists are analyzing the GC process to understand the reasons why bnAbs are difficult to produce.

#### 2.2.1. Lower BCR Affinity

Andrabi and Jardine et al. found in their study on immunogenic design for BCR that the inferred lineage BCR sequence expressing HIV bnAbs, or in the analysis of the bnAb lineage, that B cells with the smallest mutation of the common ancestor sequence had poor binding ability to HIV-1 Env, or even did not bind at all [45,46]. Neutralizing epitopes are less easily recognized by B cells than many non-neutralizing epitopes. It seems that B cells that recognize simple non-neutralizing epitopes can enhance affinity through appropriate SHM processes. For example, in one experiment, mice were immunized with commonly used nitrophenol hapten. It was found that high-affinity antibody could be formed only one week after 1–2 key amino acid mutations [47]. Due to the exposure of many non-neutralizing epitopes that are easy to identify, B cells with neutralizing epitope specificity will be at a disadvantage in antigen competition with B cells with non-neutralizing epitope specificity [43] (Figure 2). In this case, specific B cells with bnAb epitopes obtain fewer antigens, fewer Tfh cell helpers, and fewer SHM. These factors accumulate over time, making it difficult for B cells with bnAb potential to mature into B cells that truly produce bnAbs.

#### 2.2.2. Lower Frequency of Precursor B Cells

Due to the autoreactivity carried by precursor B cells with bnAb epitopes [36], their frequency in the B-cell bank is also controlled at a low level, and precursor B cells that can produce bnAbs are very rare in the B-cell bank [48,49,50]. The frequency of identifying precursor B cells with a given epitope depends on the inherent structural characteristics of the epitope. Many of these sites are not individual epitopes, but potential epitopes formed by a specific region of the antigen as a whole. More than 50% of the HIV-1 Env molecular weight is composed of glycans that mask neutralizing epitopes [51,52]; due to the protection of neutralizing epitopes by glycans, the number of precursor B cells with neutralizing potential will be much smaller, the contact angle is strictly limited, and B cells must avoid reacting with their own antigens. Joseph’s analysis of different VRC01 class (CD4-binding-site-specific) bnAb lineage-targeted immunogens on their respective precursor B cells determined that these B cells were extremely rare, with frequencies of 1 in 2.4 million human B cells [49]. PGT-121 (high-mannose-patch-specific)-like antibody precursor B cells are even rarer [53]. Precursor B cells with such a low frequency may not be able to successfully activate in vivo when recognizing the presence of other epitope B cells (Figure 2), which is also one of the reasons why bnAbs are difficult to produce.

#### 2.2.3. Less Help from Tfh Cells

In addition to the limitations of the B cells themselves, T follicular helper cells (Tfhs) also have an important impact on the B-cell response. Tfh cells can help early B cells at the T–B boundary enter GCs and help B cells complete the process of antibody affinity maturation within GCs [54,55,56]. The immune response experiments of Dal and Shih’s antibody heavy chain transgenic mice to haptens showed that low affinity initial B cells entering GCs are limited by high-affinity initial B cells [57,58,59]. This competitive relationship may be the main reason for limiting the recruitment of B cells with the potential to produce neutralizing antibodies into the GC. If there is a higher quantity or higher quality of Tfh cells (i.e., larger Tfh assistance), low-affinity initial B cells may have a higher chance of entering GCs and experiencing affinity maturation (Figure 3). The hypothesis above was supported by the relationship between memory Tfh cells and bnAb in HIV-1 infected persons [60,61]. In addition, positive associations between GC B cells and Tfh nuclei neutralizing antibodies were also found in BG505-SOSIP-immunized macaques [62]. It can be inferred from the above evidence that improving the quantity and quality of Tfh is one of the important conditions for the generation of bnAbs.

In summary, B-cell affinity, frequency, and the ability to obtain Tfh assistance are three factors that together lead to multiple restrictions on B cells that recognize neutralizing epitopes in the affinity maturation process, resulting in difficulty in producing bnAbs.

### 2.3. Attempts to Induce bnAbs

BnAbs can be found in HIV-1-infected individuals and SHIV-infected rhesus monkeys [29,63], but they cannot be induced in existing vaccination experiments on animal models or humans [64]. With a deeper understanding of the mechanism of bnAb production, researchers have attempted different strategies, hoping to induce the production of bnAbs through vaccination. Colin and Saunders are studying the affinity of precursor B cells that have changed to bnAbs, combining the analysis of these lineages with the structural analysis of antibodies and their ligands, optimizing immunogens, and testing their ability to trigger the cell lineage of bnAb B through vaccination in preclinical animal models and human clinical trials [50,65,66]. The B-cell lineage-targeted immunogen is designed to drive the antibody response by maturing rare bnAb precursor B cells [30]. Sequencing the BCR of B cells with the ability to secrete bnAb can reconstruct the process of bnAb affinity maturation and infer an unmutated common ancestor (UCA). BnAb UCA often has a low affinity with unmodified Envs, so it is commonly used for immunogenic design using surface-unmodified methods such as shortening variable loops or removing key enzymes, which is germline-targeting [48,67,68]. High-quality GC reactions require the help of Tfh cells. Some researchers try to induce bnAb by inducing sufficient Tfh [60], which can keep the cell lineage of bnAb B cells in the germinal center, or ensure that they are recycled back to the germinal center, in order to make them experience improbable mutations (these mutations are not routinely generated by somatic hypermutations, but are critical for broad neutralization [69]) that they must experience during the maturation of bnAbs. Given the importance of the quantity and quality of Tfh cells in producing effective antibody responses in natural HIV-1 infections, some scientists have attempted to improve humoral immunity by inducing or regulating Tfh reactions. Researchers have found that agonists [70] of Toll-like receptors (TLRs) 4 [71], 7/8 [72], and 9 [73], either alone or in combination, can induce stronger GC Tfh responses. PLGA (MPL+R484), a nanoparticle adjuvant containing the TLR ligand, significantly enhanced the germinal center response of mice [74] and macaques [62]. When mice are immunized with higher doses of antigens, GC Tfh and B-cell responses also increase [75]. The success of these studies is also beneficial for the development of human vaccines.

Animal models can effectively detect the success of the induction of a bnAb strategy. Studies with knockout B cells with bnAb UCA BCR into mice can preliminarily speculate on the role of a bnAb induction strategy in their amplification [76,77]. Recently, the bnAb induction model of simian human immunodeficiency virus (SHIV) infection in monkeys has provided a tool for the study of antibody environment co-evolution, and can serve as a guide for inducing bnAb [63].

## 3. Broad-Spectrum Vaccine Design to Induce bnAbs

Amino acid mutations in the receptor binding site on the surface of mutable viruses can affect antibody binding efficiency, leading to immune escape that ultimately impairs vaccine efficacy. Taking HIV-1 as an example, the env protein of HIV-1 contains important antigenic epitopes [78,79], and the mutations that occur have invariably reduced the efficacy of existing vaccines [80,81]. BnaAbs, which recognize conserved regions on the surface of the virus, can circumvent the escape brought about by a large fraction of mutations. An increasing number of bnAbs are being isolated from HIV-1-infected individuals, also implying the feasibility of relying on immune responses to induce bnAbs.

### 3.1. The Need for Broad-Spectrum Vaccine Development

The need for broad-spectrum vaccine development in the context of the widespread prevalence of mutable viruses to prevent their transmission is a major challenge. The current vaccine landscape is exemplified by SARS-CoV-2, for which all vaccines and vaccine candidates entering clinical trials are based on seven strategies [2]: protein subunit vaccines, inactivated virus vaccines, live attenuated vaccines, DNA vaccines, mRNA vaccines, virus vectored vaccines, and virus-like particle (VLP) vaccines. These vaccines attempt in some ways to induce bnAbs, but they are not yet sufficient to address the challenges posed by immune escape from mutated strains. The control of influenza awaits the birth of a broad-spectrum vaccine in the “strain-a-year, needle-a-year” model, and no vaccine has been available for HIV-1 since its discovery until 40 years ago. Addressing the issue of variant virus prevalence and the resulting immune escape of mutant strains will require unfolding explorations into the mechanisms underlying bnAb generation.

### 3.2. Strategies for Inducing bnAb Can Guide Vaccine Design

The generation of bnAbs is inseparable from the robust T cell and B cell responses in the GC, and vaccines may be able to induce bnAbs if they address the multiple limitations that B cells have in neutralizing epitopes during affinity maturation. How low-affinity naïve B cells are pulled into the GC and undergo the process of affinity maturation with the help of Tfh cells is a central question for broad-spectrum vaccine design. As insight into immune mechanisms has grown immensely, investigators have identified factors that can invest in the development of broad-spectrum vaccines for mutable viruses by eliciting robust and long-lasting GC responses, lending impetus to the development of broad-spectrum vaccines.

#### 3.2.1. Role of TLR Signaling during bnAb Generation

In the 1980s, Janeway et al. speculated that the immune system would have specific pattern recognition receptors (PRRS) to recognize the relevant molecular patterns (PAMPs) of different antigens [82]. Further verification of this hypothesis was provided by Janeway, who found TLRs that can initiate the activation function of antigen-presenting cells [83]. Under conditions in which scholars generally believe that the PRR-mediated immune response achieves innate immunity, the specific response of B cells to lipopolysaccharide (LPS) drives the discovery that B cells express TLRs [84]. Following the discovery that TLR stimulation enhances T-cell-dependent as well as T-cell-independent antibody responses following the recognition of LPS and other bacterial cell wall components and the recognition of pathogen-derived nucleic acids [85,86], these findings spawned further studies between TLRs and antibody production. TLR7 binds and responds to ssRNA in the endosomal compartment, and given that retroviruses have ssRNA genomes, the role of TLR7 in retroviral infection has received considerable attention. Anthony et al. showed that in mice, TLR7 expressed within B cells can enhance GC IgG responses [87]; genetic studies in humans have also pointed to a promotional role for TLR7 in immune responses elicited by HIV-1 infection [88], and the weak activating effect of HIV-1 on TLR7 may also contribute to its difficulty in being neutralized [89]. This suggests that the role played by TLR7 in immune responses elicited by viral infection may be a key to the development of broad-spectrum vaccines. On this basis, Hong found that Q β- VLP can activate initial CD4+T cells by stimulating TLR7 within B cells with internal nucleic acids [90]. This T–B homologous interaction may be beneficial for inducing strong GC responses. Then, Guo selected the antigen form of phage VLP with internal nucleic acid encapsulation, and covalently linked it with exogenous antigen proteins for the development of a SARS-CoV-2 vaccine based on this [91]. Guo chose to covalently connect AP205 with SARS-CoV-2 RBD, and constructed AP205-RBD fusion protein particles. The preliminary immune results showed that it can cause a relatively persistent GC reaction and also have a certain neutralization effect on SARS-CoV-2 variations. As a result of the characteristic of delivering mRNA into cells, mRNA vaccines may also stimulate the B-cell-intrinsic TLR7, which may be the reason for its success against COVID-19.

#### 3.2.2. Tfh- and B-Cell Interactions

In general, Tfh cells depend on B cells, while the differentiation of immature B cells into GC B cells and plasma cell (PCs) depends on Tfh cells [92], and the proper regulation of humoral immunity also depends on these feedback circuits. Tfh cells migrate to the T–B boundary during early differentiation and interact with antigen-specific B cells [93]. In addition to controlling the selection of high-affinity GC B cells, GC Tfh cells are also key powerhouses for GC B-cell differentiation into memory B cells and PCs to shape long-term humoral immunity. HIV-1 infection has been shown to reduce the normal tolerance control that restricts the development and maturation of polyreactive or autoreactive B cells, thereby creating a relaxed immune environment for the development of bnAbs [60,94]. The strategy of inducing bnAb by the HIV-1 vaccine in HIV-1-negative individuals must be able to reconstruct the allowable immune microenvironment that occurs during HIV-1 infection as safely as possible. One method is to induce a large number of Tfh cells. Raising the number of Tfh cells may help to maximally recruit B cells, and thereby reduce competition for Tfh; this can allow low-affinity neutralizing epitope-specific B cells to undergo clonal proliferation, become memory cells, and be awakened in a secondary immune response. To obtain sufficient antigen affinity, B cell clones with neutralizing specificity require the help of different types of Tfh cells. Meanwhile, a certain number of high-quality Tfh cells may also be the optimal condition for antibody affinity maturation targeting neutralizing epitopes.

The main function of Tfh cells is to resist viruses. The experimental results obtained in mice, non-human primates (NHP), and humans have confirmed that Tfh cells are crucial for the antibody response [54]. Studies on SIV and SHIV infections in rhesus monkeys (RMS) have shown that the formation of broad-spectrum neutralizing antibodies requires some assistance from Tfh cells. This is the first model system that can directly detect GC Tfh cells from lymphatic tissue. During chronic SIV infection, the GC Tfh cells and GC B cells in RMS gradually increase [95,96,97], and the RMS individuals who produce the highest SIV env-specific antibody titer also have the highest frequency of GC Tfh cells. The important role of Tfh cells in the vaccine immune response is similar to their role in immune response induced by viral infection [98]. To simulate the immune microenvironment of HIV-1-infected individuals in the regulation of protein immunity, the use of adjuvants is also a method to regulate the differentiation of Tfh cells and the induction of GC B cells. Alum stimulates Tfh cell responses, but is relatively less effective [99]. Oil in water adjuvant, represented by MF59, is also used in human vaccines, which can induce CD4+T cell responses dominated by Tfh cells through IL-6 stimulation of BCL6 expression [100]. The main mechanism of action of NKT adjuvant may be that NKT cells provide early cytokines to B cells before IL-4 and IL-21 are provided by Tfh cells. Gaya et al. found that NKT KO mice had a reduced proportion of GC B cells following influenza virus infection, also illustrating that early IL-4 is necessary for B-cell responses [101]. These cytokines can all actively promote Tfh cells to help B cells, opening potential avenues for the development of novel adjuvants.

Moreover, mRNA vaccines have also been found to induce strong Tfh responses, suggesting that this may be a contribution from the adjuvant effect of LNP itself. Therefore, ionizable LNP is considered as an adjuvant in combination with protein vaccines to promote the assistance of Tfh cells [94,102].

#### 3.2.3. B Cell Linage Vaccine Design

B-cell lineage vaccine can bring precursor B cells with the potential to produce bnAbs into the GC and eliminate the interference of non-neutralizing epitope-specific B cells as much as possible. In influenza, due to the relatively conservative stalk domain of hemagglutinin (HA), researchers have used several methods to drive the entry of naïve B cells that recognize HA stalks into the GC, such as high glycosylated HA [103] and the construction of headless HA [104,105,106]. This strategy has achieved certain results in animals [105,106]. In HIV-1, Jardine designed a B-cell lineage targeting immunogen eOD-GT8 [48], which has sufficient affinity for the inferred UCA of VRC01 [107,108], and also has a certain degree of binding with the UCA of other VRC01 class bnAbs. eOD-GT8 has shown the ability to amplify VRC01 precursor B cells in mice, macaques [109], and even humans [110], indicating that it is feasible to drive bnAb generation by maturing scarce bnAb precursor B cells.

#### 3.2.4. Prolonged Antigen Stimulation Can Enhance GC Responses

Achieving a sustained, slow delivery of antigens is another method for enhancing the Tfh response, which is reasonable and feasible in vaccines. The sustained, slow delivery of antigens can better simulate natural infections. Recent studies have shown that antigen-controlled release strategies are feasible, and the slow and sustained release of antigens over a longer period of time may induce stronger immune responses than traditional immune models [111,112,113]. Compared with NHPs receiving routine immunization, the autologous HIV-1 bnAbs induced by osmotic pump immunization against HIV-1env in NHPs after two weeks significantly increased [112]. The development of HIV-1 bnAb in animals immunized with osmotic pumps is also faster than in animals immunized with traditional methods, suggesting that prolonged antigen release may accelerate the production of GC B cells with a high affinity for HIV-1 env. Osmotic pump-immunized NHPs had higher frequencies of GC Tfh cells, and these GC tfhs also had higher Ki67 positivity. In addition, the slow delivery of antigens can also enhance the stability of intact antigenic proteins. For unstable viral antigens, such as HIV-1, after the GC response reaches its peak in routine immunity, most of the antigens presented by follicular dendritic cells (FDCs) to GC B cells may become non-natural proteins and degradation products. This may expose antigen epitopes, which are usually hidden or not present in the natural form of antigen proteins of the virus [114]. Antigen slow delivery systems can preserve the integrity of FDC recognition of viral antigens. Modulating antigen and adjuvant kinetics can improve vaccine efficacy, and insight into these processes will provide considerable guidance for vaccine design.

#### 3.2.5. Multivalent Antigens Can Enhance the Broad Spectrum of Antibodies

Antigen multivalency was strongly correlated with higher antibody titers. For hepatitis B virus [115] and human papillomavirus vaccines [116], the multivalency of virus-like particles (VLPs) is seen as a key element in the success of these vaccine antigens [117]. Vaccine design strategies for antigenic multivalency have also been extensively attempted in HIV-1 and SARS-CoV-2. The Ad26 mosaic vaccine program developed by Janssen et al. combined virally vectored protein enhancers and immunogen sequences; these vaccines were optimized in an attempt to address global HIV-1 diversity, and were demonstrated to slightly improve antibody quality [118]. The above results suggest that increasing the surface multivalency of antigens has a positive effect on bNAb induction. In the context of the emergence of different variants in the SARS-CoV-2 pandemic, researchers explored the impact of sequential immunization strategies on broad-spectrum antibodies. Juan assessed the neutralizing activity and protection afforded by the BA1-S subunit vaccine when administered with wild-type S protein (WT-S) in combination or used as a booster dose. Compared with the WT-S protein alone, the WT-S/BA1-S mixture or WT-S anise and BA1-S enhancer induced neutralizing antibodies against the pseudotype Omicron BA1, BA2, BA2.12.1, and BA5 variants and similar or higher neutralizing antibodies against the original SARS-CoV-2 [119].

Antigen effective titers also have a direct effect on B-cell breadth, and by changing the affinity range of B cells activated after immunization in response to multivalent immunogens, the magnitude of GC and PC responses also increases with increasing antigen-effective titers. Recently, Shane et al. examined the immune responses induced by engineered protein immunogens targeting different valencies of the B-cell lineage, confirming the multifaceted impact of antigenic valency on the composition and differentiation of B-cell responses in vivo in terms of immunological mechanisms; this provides theoretical support for the strategy of inducing bnAbs by multivalent antigens [120]. Given that the magnitude and quality of the B-cell response depends on the number of antigenic titers that multivalent antigens can reliably induce in desired variant viral antibodies, the ability to precisely quantify and control effective titers in vivo should also be an ability that a broad-spectrum vaccine should possess.

## 4. Humoral Immunological Memory

A successful humoral immune response has two key characteristics. The first characteristic is the affinity and broad spectrum of antibodies secreted by PCs; the second feature is the reactive humoral immunological memory. Memory B cells quickly differentiate into antibody-producing plasma cells to respond to reinfection when defending against reinfection by homologous viruses. At the same time, some memory B cells are driven to re-enter the GC to achieve multiple rounds of SHM. It is also a potential attempt in broad-spectrum vaccine design to induce memory B cells to accumulate and produce SHMs after virus infection or immunization.

During the GC reaction, B cells with lower affinity will become memory B cells earlier [121,122] and exit the GC. The vaccine design strategy that drives memory B cells back into GCs is very persuasive. This can enable B cells to further obtain more extensive mutations needed to neutralize certain viruses, which is unlikely to occur in routine immunization. Unfortunately, there is currently very little research in this area.

However, in the context of strong viral mutation ability, memory B cells sometimes play a counter role in the production of bnAbs. There is already evidence that in SARS-CoV-2 or influenza virus infections, the earliest infected strain can elicit an antigenic ‘imprint’. This “imprint” will continue to be enhanced by subsequent mutant strains [123], which is detrimental to the production of bnAbs [124], and may concentrate the produced antibodies on mutable epitopes [125]. This concept is called the antigen original sin (OAS) [126]. In the receptor binding domain of influenza, the “head” is more likely to induce immune activation than the “stalk” [127]. Although the “stalk” epitope is more conserved, the preexisting “head” specific memory will induce more antibodies targeting the “head” in the secondary reaction [128]. Therefore, overcoming the obstacles brought by the OAS is also an important step in the design of broad-spectrum vaccines. More in-depth research on memory B cells will provide great guidance for vaccine design.

## 5. Perspectives on Broad-Spectrum Vaccine Design

Whereas existing vaccines are based on prior experience and repeatedly fumbled, bNAb generation requires robust and long-lasting GC responses, and the birth of a broad-spectrum vaccine requires establishing a link between immunogens and GC responses. After factors influencing the GC response were gradually uncovered, investigators began trying to exploit the GC reaction law to invest in vaccine development. To achieve long-lasting GC reactions, investigators mostly choose to achieve better activation of DC cells using traditional soluble protein antigens plus TLR agonists [129,130,131], or to achieve sustained presentation of antigens using a slow-release approach [132]. However, the broad spectrum of vaccines remains unsatisfactory, and understanding the role that mechanisms of bNAb generation actually play in vaccine applications may require intensive exploration and attempts to enable researchers to gain a greater understanding of immune mechanisms in vaccines to design truly well-established broad-spectrum vaccines against mutable viruses.

Facing the challenge of various major infectious diseases, the development of innovative vaccines not only relies on incremental improvements in materials, expression, and delivery methods, but also requires deeper exploration of the basic theory of immunity. The study of the mechanism of bnAb generation is a meridian for the design of broad-spectrum vaccines for mutable viruses, and deeper mining of the mechanism of bnAb generation can provide a reference for the design of broad-spectrum vaccines for mutable viruses.

## Figures and Tables

**Figure 1 vaccines-11-01143-f001:**
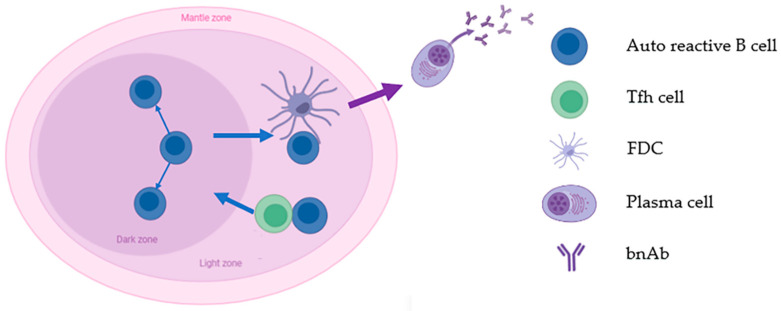
Some B cells with autoreactivity continuously experience SHM in the germinal center. This process is necessary for the formation of plasma cells that can produce broad-spectrum neutralizing antibodies.

**Figure 2 vaccines-11-01143-f002:**
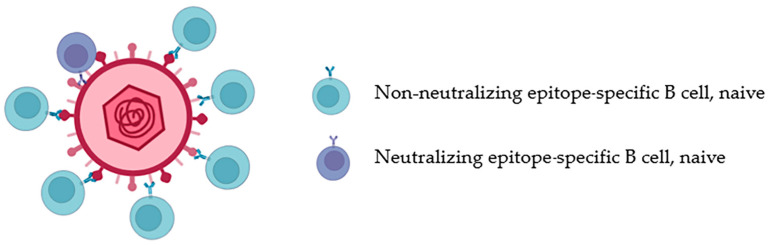
Since neutralizing epitopes (the stalk of the red part) may be hidden by non-neutralizing epitopes (the head of the red part), neutralizing epitope-specific B cells are more difficult than their non-neutralizing epitope-specific B Cell–cell recognition antigens, and their numbers are also far lower than those of non-neutralizing epitope-specific B cells.

**Figure 3 vaccines-11-01143-f003:**
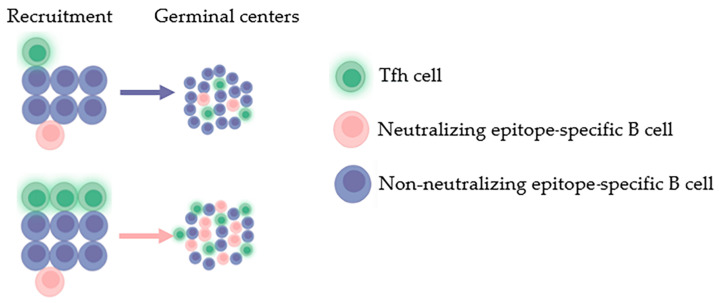
Due to the low BCR affinity and relatively low frequency of neutralizing epitope-specific B cells, their competition with non-neutralizing epitope B cells for limited Tfh cells is limited by the latter. If there are more Tfh cells, it will increase the opportunity for neutralizing epitope-specific B cells to be assisted by Tfh cells, thereby enabling more neutralizing epitope-specific B cells to enter the GC and undergo subsequent affinity maturation processes.

**Table 1 vaccines-11-01143-t001:** Seven clinically tested HIV vaccines that induce Humoral immunity.

Trial	Start	End	Vaccine	Location	Efficacy	References
VAX003 (NCT00006327)	1999	2000	Bivalent CRF_01AE/B gp120 in alum	Thailand	No efficacy	[16,17]
VAX004 (NCT00002441)	1999	2000	Bivalent clade B gp120 in alum	United States, Europe	No efficacy	[18,19]
RV144 (NCT00223080)	2005	2009	ALVAC with gag/pro/Env; bivalent CRF_01AE/B gp120 in alum	Thailand	Estimated 60% vaccine efficacy at 12 months; 42-month efficacy, 31.2%	[20,21,22]
HVTN 505 (NCT00865566)	2009	2017	DNAs with clade B gag/pol/nef and DNAs with clade A, B, C Envs; adenovirus type 5 with gag/pol and clade A, B, C Envs	United States	No efficacy	[23,24]
HVTN 702 Uhambo (NCT02968849)	2016	2021	ALVAC-C with gag/pol/Env; bivalent gp120s in MF59	South Africa	No efficacy	[25]
HVTN 705 Imbokodo (NCT03060629)	2017	2021	Ad26, 4 valent T cell mosaic genes, boost with clade C gp140 Env	Sub-Saharan Africa	No efficacy	[26]
HVTN 706 Mosaico (NCT03964415)	2019	On going (est. 2024)	Ad26, 4 valent T cell mosaic genes, boost with clade C gp140 Env+B cell mosaic gp140 Env	United States, Spain, Central/South America	Ongoing	[27]

## Data Availability

No data.

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
