# Peer review of "The Mechanism of bnAb Production and Its Application in Mutable Virus Broad-Spectrum Vaccines: Inspiration from HIV-1 Broad Neutralization Research"

_vaccines, 2023, doi:10.3390/vaccines11071143_

Round 1
Reviewer 1 Report
- Major comments:
In this study, Xinyu Zhang et al. reviewed the mechanism by which the human immune system produces broad neutralizing antibodies and its induction strategies, which is helpful in the design process of broad-spectrum vaccines.
- General concept comments
Here are some considerations for the study:
1. The study provides a good analysis of broadly neutralizing antibodies and provides some suggestions for broad-spectrum vaccine design. However, the evidence is not so strong or clear. For better illustration, please summarize in a Figure showing the ways to generate broadly neutralizing antibodies.
2. Please tabulate the recent advances of broadly neutralizing antibodies (bnAb) found against viruses like HIV, IAV, SARS, or SARS-CoV-2, etc.
3. Please tabulate the recent advances of broad neutralizing vaccines developed against viruses like HIV, IAV, SARS, or SARS-CoV-2, etc.
- Specific comments:
1) Typo of “1. Boardly neutralizing antibody”.
2) Page 6, “RSM” or “RMS”?
3) The full name of “FDCs” should be noted on page 6.

Fine.
Reviewer 2 Report
The authors intend to summarize the studies on broad-neutralizing antibodies and broad-spectrum vaccines.
Several suggestions:
1. Please define the [broad neutralizing antibodies]. If it is the same definition as that written in [Nature. 2022 Feb;602(7896):314-320. Broadly neutralizing antibodies target a haemagglutinin anchor epitope], then the [broad neutralizing antibodies] may not be induced through [2.2.4. Multivalent antigens can enhance the broad spectrum of antibodies].
2. It is better to have figures in a review article to help readers understand the content.
3. [veriations] were written in the entire manuscript. [variations]?
4. Line 4 in [Introduction], [positive-stranded]? suggested to be [positive-sense].
5. Please use the same term: [Tfh], [TFH], and others.
6. [variant viruses] are used in the entire manuscript. [viral variants]?
7. No line number in this submission.
Reviewer 3 Report
This manuscript seems scope on the people who live with HIV-1, but the introduction and scope of this manuscript were not clear. However, the content is interesting and very informative.
Suggest re-writing again with a more comprehensive and informative introduction, especially the introduction. The review article must be high information and quality, otherwise, this manuscript is similar to the narrative review.
Major concerns.
1. The title of this manuscript is too broad and does not mention HIV, which was the main scope.
2. The Introduction was insufficient information focusing on HIV.
Suggest adding more paragraphs about HIV related to the scope of the manuscript.
3. This manuscript's scope is on the HIV subject. But the comparison with the healthy control required describing the similarities or differences.
Suggest checking the typos in the manuscript.
Reviewer 4 Report
Please refer to the attached document.
In this comprehensive review, the author provides a detailed examination of the definition of broadly neutralizing antibodies (bnAbs), the underlying mechanisms of their induction, and relevant factors influencing bnAb development, including the coordinated interplay among germinal center (GC) cells, T follicular helper (Tf) cells, and B cells. The review further summarizes and introduces strategies for bnAb induction, with a specific focus on a key aspect of broad-spectrum vaccine design—the mechanism underlying bnAb induction, which holds insights for researchers in the field. However, there is room for improvement in the overall logical structure of the article. There is a slight overlap between the first and second parts, and the content would benefit from further adjustments and divisions to effectively present the concept of bnAbs, their specific mechanisms of generation, and potential influencing factors. Building upon this foundation, the review could delve deeper into elucidating the mechanisms driving bnAb induction to facilitate the design of current broad-spectrum vaccines or explore how known mechanisms can inform the development of broad-spectrum vaccines. Additionally, while the preface and abstract make a mention of SARS-CoV-2 as an example, the primary focus of the article is centered around HIV. Therefore, it is advisable to downplay the foreshadowing of SARS-CoV-2 to enhance the overall coherence and cohesion of the review.
Other comments:
1. Given that the elucidation of the induction mechanism of broadly neutralizing antibodies (bnAbs) serves as the primary focus of this article, it is advisable to supplement the textual explanations with illustrative figures to enhance the clarity and comprehension of the relevant mechanisms.
2. Page 1: The sentence “There have been many successful vaccines in human history, such as smallpox, which have been optimized and developed based on previous experience.” "Based on previous experience" requires further elucidation to enhance clarity and precision.
3. Page 2: The assertion made in this sentence “it is difficult to develop a mature broad-spectrum HIV-1 vaccine without immunological understanding of the selective promotion of bnAbs development by CD4+T cells and B cells.” regarding the absolute correlation between the understanding of immunology and the development of a broad-spectrum HIV vaccine is overly definitive. It is recommended to moderate the statement to acknowledge that a comprehensive understanding of immunology can significantly contribute to the advancement of HIV broad-spectrum vaccine research.
4. Page 3, Part 1.2.2: In the sentence "PGT-121 like antibody precursor B cells are even rarer.", the specific classification of antibodies, particularly the type to which the PGT-121 antibody belongs, requires clarification and definition for improved comprehension.
5. Page 3, Part 1.2.2: In this section, the author relies heavily on conclusive language and speculative statements without adequate references to substantiate the accuracy and scientific rigor of the assertions. It is recommended to incorporate relevant citations to support the author's claims and enhance the scholarly credibility of the article. Additionally, it is advisable to prioritize the utilization of research data over general language overviews or speculative discussions to strengthen the evidential basis of the arguments presented.
6. Page 4: What does this sentence mean?
“In summary, due to the combination of B-cell affinity, frequency, the ability to obtain Tfh assistance is determined by three factors…”
7.Page 4, “The B-cell lineage vaccine is designed to drive the antibody response by maturing rare bnAb precursor B cells [33].”: Kindly provide a concise introduction to " The B-cell lineage vaccine " that emphasizes its design to facilitate the maturation of rare precursor B cells for broadly neutralizing antibodies (bnAbs).
8. Page 4, Part 1.3. Attempts to induce bnAbs: The author provides a concise overview of the induction mechanisms, specific design methodologies, and current progress in generating broadly neutralizing antibodies (bnAbs). However, to ensure the comprehensive coverage of the subject matter, it is recommended to expand the review by encompassing the broader scope of bnAb induction mechanisms, summarizing the current landscape of bnAb design strategies, and offering an extensive reference for researchers interested in bnAb induction.
9. Page 4: Is there adequate empirical evidence available to substantiate the assertion put forth in this statement “None of these vaccines have undergone engineering attempts to induce bnAbs and are insufficient to address the challenges posed by immune escape of the mutant strains.”? Given the significance and complexity of vaccine research, it is advisable to approach the topic with caution and avoid making overly definitive claims in the absence of a comprehensive understanding of the current state of research in the field.
10. Please provide a reference citation for the following sentence to support the scholarly basis and verifiability of the information presented:
“The concept of broad neutralizing antibody (bnAb)…antibodies with a wide neutralization width against different HIV-1 strain.” (Page 1)
“Among HIV-1-infected individuals, there are … BnAb is selected from elite control individuals and is the main antibody that exerts antiviral effects in the body.”
“Shortly after the isolation of the first batch of bnAbs…and self-reactions with human lipids and proteins.” (Page 2)
“Researchers believe that bnAbs are produced by mature memory B cells of HIV-1- infected people who have not progressed for a long time.”
“Due to the exposure of many nonneutralizing epitopes that are easy to identify, B cells with neutralizing epitope specificity will be at a disadvantage in antigen competition with B cells with nonneutralizing epitope specificity.” (Page 3)

Minor editing of the English language is necessary for the sentence, as indicated in the comments above. Please refer to the provided comments for further details.
Round 2
Reviewer 1 Report
I think that the manuscript has been improved, and the authors have addressed most of my concerns.
Fine.
Author Response
Thank you very much for your valuable comments on this manuscript.
Reviewer 2 Report
The issues I raised previously have been addressed in this revised manuscript.
Author Response

(The authors gave the same response as above.)

Reviewer 3 Report
That is acceptable for the substantial revision to make your manuscript clear and more information to the manuscript's scope.
Author Response

(The authors gave the same response as above.)

Reviewer 4 Report
The author has addressed the concerns raised in the first round of review and made significant improvements to the manuscript. The content has been enriched, and the inclusion of visual aids has enhanced the overall readability of the article. However, there are a few remaining minor issues that the author is encouraged to address further:
1. Considering that the added figures in the article provide relatively similar information, it is recommended to merge Figure 2 and Figure 3 into a single figure, ensuring a logical and concise representation. Furthermore, Figure 5 appears to lack specific content and is thus suggested for deletion.
2. Consistency in the formatting of figure captions should be ensured, using either "Fig. X" or "Figure X" consistently throughou the manuscript.
3. The meaning represented by the red icons in Figure 2 and Figure 3 should be appropriately explained in the figure legends. Please provide clear and concise descriptions for better comprehension.
4. The current resolution and definition of the figures are insufficient. It is advised to replace them with high-definition images to ensure clarity and enhance the visual impact.
Addressing these suggestions will further enhance the quality and clarity of the manuscript, providing readers with a more comprehensive and visually appealing presentation of the research findings.
